# Porcine Circovirus Modulates Swine Influenza Virus Replication in Pig Tracheal Epithelial Cells and Porcine Alveolar Macrophages

**DOI:** 10.3390/v15051207

**Published:** 2023-05-20

**Authors:** Yaima Burgher Pulgaron, Chantale Provost, Marie-Jeanne Pesant, Carl A. Gagnon

**Affiliations:** 1Swine and Poultry Infectious Diseases Research Center (CRIPA-FRQ), Faculté de Médecine Vétérinaire, Université de Montréal, Saint-Hyacinthe, QC J2S 2M2, Canada; yaima.burgher@umontreal.ca (Y.B.P.); marie-jeanne.pesant@umontreal.ca (M.-J.P.); 2Molecular Diagnostic Laboratory, Centre de Diagnostic Vétérinaire de l’Université de Montréal (CDVUM), Saint-Hyacinthe, QC J2S 2M2, Canada; chantale.provost@inesss.qc.ca

**Keywords:** porcine circovirus, swine influenza A virus, epithelial cells, macrophages, co-infection, viral pathogenesis, virus replication

## Abstract

The pathogenesis of porcine circovirus type 2b (PCV2b) and swine influenza A virus (SwIV) during co-infection in swine respiratory cells is poorly understood. To elucidate the impact of PCV2b/SwIV co-infection, newborn porcine tracheal epithelial cells (NPTr) and immortalized porcine alveolar macrophages (iPAM 3D4/21) were co-infected with PCV2b and SwIV (H1N1 or H3N2 genotype). Viral replication, cell viability and cytokine mRNA expression were determined and compared between single-infected and co-infected cells. Finally, 3′mRNA sequencing was performed to identify the modulation of gene expression and cellular pathways in co-infected cells. It was found that PCV2b significantly decreased or improved SwIV replication in co-infected NPTr and iPAM 3D4/21 cells, respectively, compared to single-infected cells. Interestingly, PCV2b/SwIV co-infection synergistically up-regulated IFN expression in NPTr cells, whereas in iPAM 3D4/21 cells, PCV2b impaired the SwIV IFN induced response, both correlating with SwIV replication modulation. RNA-sequencing analyses revealed that the modulation of gene expression and enriched cellular pathways during PCV2b/SwIV H1N1 co-infection is regulated in a cell-type-dependent manner. This study revealed different outcomes of PCV2b/SwIV co-infection in porcine epithelial cells and macrophages and provides new insights on porcine viral co-infections pathogenesis.

## 1. Introduction

The porcine respiratory disease complex (PRDC) causes enormous economic loss to the swine industry worldwide [1]. Porcine reproductive and respiratory syndrome virus (PRRSV), porcine circovirus type 2 (PCV2) and swine influenza A virus (SwIV) are the main viruses associated with PRDC and are involved in costly outbreaks in swine production sites [2].

PCV2 is a non-enveloped single-stranded circular DNA virus that belongs to the *Circoviridae* viral family, genus *Circovirus* [3]. PCV2 is the etiological agent of several diseases and syndromes such as post-weaning multisystemic wasting syndrome (PMWS), porcine dermatitis and nephropathy syndrome (PDNS), among others which are collectively named porcine circovirus-associated disease (PCVAD) [4]. The prevalence of PCV2 infection is high in swine farms all over the world [5]. However, most PCV2 infections are reported to be subclinical [5]. It is noteworthy that the virus can alter the innate immune response and cause immunosuppression, which favors co-infection and/or secondary infection with other pathogens [6]. Eight genotypes of PCV2 have been described to date (PCV2a-PCV2h) [7,8,9]. The emergence of the PCV2b genotype in Canada was associated with a significant death rate increase during the post-weaning multisystemic wasting syndrome (PMWS) outbreak in late 2004 [10] and was reported as the predominant genotype identified during the same period in samples submitted to the diagnostic service of the Faculty of Veterinary Medicine (Faculté de médecine vétérinaire) of the University of Montreal [10,11].

Swine influenza A viruses (SwIV) are enveloped single-stranded negative sense RNA viruses belonging to the *Orthomyxoviridae* viral family, genus *Alphainfluenzavirus* [3]. The subtypes of influenza A virus (IAV) are determined by the hemagglutinin and neuraminidase proteins which are embedded into the envelope of the virion. Eighteen HA subtypes (H1–H18) and eleven NA subtypes (N1–N11) have been reported to date [12,13]. SwIV subtypes H1N1, H1N2, H3N2, and A(H1N1)pdm09 have been the most frequently reported influenza A virus subtypes in swine worldwide [14]. In Canada, SwIV subtypes H1N1, H1N2 and H3N2 have been circulating in swine herds since at least the 1980s [15,16,17,18,19,20,21]. In 2009, A(H1N1)pdm09 was reported in swine in the Canadian provinces Manitoba, Alberta, Saskatchewan and Quebec [22]. Infections caused by SwIV are characterized by high morbidity and very low mortality [23]. It is known that IAV induces high levels of cytokine release very often associated with viral loads and severity of the disease [24].

PCV2 co-infections with SwIV or PRRSV have been extensively studied in vivo [25,26,27,28,29,30,31,32,33] and in vitro [34,35,36,37]. Even though PCV2 and SwIV co-infections are also prevalent in pigs [1], pathogenesis studies concerning this co-infection are scarce. Interestingly, an in vivo epidemiologic assessment showed that PCV2-positive pigs were more likely to be infected with SwIV than PCV2-negative pigs [38]. Other authors have reported that PCV2/SwIV H1N1 co-infection did not affect PCV2 replication in experimentally infected pigs, whereas PCV2 infection did increase SwIV-related clinical disease in the infected animals [39].

PCV mostly targets cells of monocyte/macrophage lineage in vivo but can infect other cell types [40,41]. SwIV preferentially targets epithelial cells of the respiratory tract, but alveolar macrophages can also be infected [42,43,44]. Epithelial cells detect viral pathogens via pattern recognition receptors (PRRs) and release anti/proinflammatory cytokines to recruit and activate innate immune cells, which finally trigger innate and adaptive immune responses [45,46]. On the other hand, the importance of macrophages for host defense against invading pathogens is well-known [47]. They are professional phagocytic cells that clear infectious particles and apoptotic cells and participate in the adaptive immune response by T cells activation via antigen presentation [47]. It is well-known that concomitant infections modulate viral pathogenesis and affect host immune cells’ responses [1]. However, there is a lack of information regarding the effect of PCV2b/SwIV co-infections at the cellular and molecular level. The objective of the present study was to evaluate the effects of PCV2b/SwIV co-infections on virus-targeted host cells such as swine respiratory epithelial cells and porcine alveolar macrophages. The results of the present study revealed that PCV2b modulates SwIV replication during co-infection, meanwhile affecting SwIV mRNA cytokine expression and cellular genes modulation in infected cells, in a cell-type-dependent manner.

## 2. Materials and Methods

### 2.1. Cells

Newborn pig tracheal epithelial cell line (NPTr) and immortalized porcine alveolar macrophage (iPAM 3D4/21) cell line were used for all PCV2b and SwIV single and co-infections. Madin–Darby Canine Kidney (MDCK) cell line was used for SwIV titration and propagation. The NPTr cell line was kindly provided by Dr. M. Ferrari (Instituto Zooprofilattico Sperimental, Brescia, Italy) [48]. The NPTr and the MDCK (ATCC CCL-34) cells were cultured in Eagle’s Minimum Essential Medium (EMEM) (Wisent Bioproducts, Saint-Jean-Baptiste, QC, Canada) supplemented with 10% fetal bovine serum (FBS) (Wisent Bioproducts, Saint-Jean-Baptiste, QC, Canada), 1 mM sodium pyruvate, 10 I.U./mL of penicillin, 10 μg/mL of streptomycin and 250 g/L amphotericin B solution (Wisent Bioproducts, Saint-Jean-Baptiste, QC, Canada) [48]. The iPAM 3D4/21 cell line (ATCC CRL-2843) was maintained in RPMI 1640 medium (Invitrogen Corporation, GibcoBRL, Burlington, ON, Canada) with 10% FBS and adjusted to contain 2 mM L-glutamine (Invitrogen), 10 mM HEPES (Invitrogen), 1 mM sodium pyruvate, 1 mM non-essential amino acids (Invitrogen) and 0.1 mg/mL streptomycin/100U penicillin solution (Invitrogen). All cells were cultivated in a humidified incubator in 5% CO_2_ atmosphere at 37 °C. 

### 2.2. Viruses

The PCV2b strain (FMV-06-0732) was isolated from a clinical case in Quebec in 2006 (GenBank accession number: JQ994270) [10]. It was serially propagated in NPTr cells and then purified and concentrated following ultracentrifugation on a 30% sucrose cushion using the SW28 Beckman Coulter rotor (Beckman Coulter Canada Inc., Mississauga, ON, Canada) at 25,000 rpm for 4 h. 

The SwIV H1N1 (A/swine/St-Hyacinthe/148/1990) was isolated from pigs with respiratory symptoms during a 1990/91 outbreak of respiratory disease in Quebec (GenBank accession number: U11703) [16]. The SwIV H3N2 (A/swine/Quebec/1708732/2015(H3N2) was isolated from a pig with acute respiratory disease (GenBank accession number: KX571068, KX571087, KX571114, KX571138, KX571161, KX571164, KX571200, KX571217). The sample was submitted by a veterinarian practitioner to Diagnostic Veterinary Virology Laboratory (DVVL) of the University of Montreal in 2015 as part of routine swine flu diagnostic testing. Viral titers of PCV2b and SwIV stocks were determined in NPTr cells and MDCK, respectively, via the Spearman–Kärber method [49,50]. Viral titers were expressed in tissue culture infectious dose 50% per mL (TCID_50_/mL). 

### 2.3. Immunofluorescence Assay

Single infection and co-infection were confirmed using an immunofluorescence assay (IFA) as previously described [51]. Briefly, the cell medium was removed and infected cells were fixed with a mixture of acetone–methanol (50/50, *v*/*v*) and incubated for a 20 min period. Thereafter, fixed infected cells were washed three times with a PBS solution and then permeabilized with a solution containing 0.1% triton X-100 in PBS and incubated for 10 min. After incubation with a blocking solution (1% bovine serum in PBS-Tween for 20 min), cells were incubated with a 1/200 dilution of the polyclonal PCV2 porcine antiserum [51] at 37 °C for 90 min and/or with 1/200 diluted monoclonal mouse anti-NP SwIV primary antibody (Bio-Rad, Hercules, CA, USA). Then, the cells were washed three times with a solution containing 1% bovine serum in PBS-T and incubated with a 1/75 dilution of a goat anti-swine rhodamine conjugated secondary antibody (Jackson ImmunoResearch, West Grove, PA, USA) and/or a 1/200 dilution of goat anti-mouse FITC conjugated secondary antibody (Invitrogen Corporation, GibcoBRL, Burlington, ON, Canada ), at 37 °C for 60 min. After three washing steps, the cells were visualized using a Leica DMI 4000 inverted widefield fluorescence microscope (Leica Microsystems Inc., Richmond Hill, ON, Canada). Pictures were acquired with a DFC 490 digital camera (Leica Microsystems Inc.) and images were analyzed using Leica Application Suite Software, version 2.4.0 (Leica Microsystems Inc.) and ImageJ software (Laboratory for Optical and Computational Instrumentation, LOCI, University of Wisconsin, Madison, WI, USA) [52]. Mock-infected cells were included in the immunofluorescence assay to verify the specificity of the antibodies (Appendix A).

### 2.4. PCV2b/SwIV Co-Infection 

To obtain PCV2b/SwIV co-infected cells, NPTr or iPAM 3D4/21 cells were firstly infected with PCV2b at an MOI of 0.05. Afterwards, PCV2b-infected cells were passaged at least three times in the presence of the virus to achieve a stable persistent infection. PCV2b-infected cells were thereafter dispensed into 24-well plates (1 × 10^5^ cells/well) and then co-infected with SwIV H1N1 or H3N2 subtypes at a MOI of 1 for 1 h in the presence of trypsine (1ug/mL). Cells were then washed three times with PBS. Fresh medium with 2% FBS was added to the wells and the plates were incubated in a humidified incubator with 5% CO_2_ at 37 °C. After 4, 12, 24, 48 and 72 h SwIV following infection, the cells were subjected to 2–3 rounds of freeze–thaw cycles to release virus particles and cell debris, which were then removed via centrifugation at 8000 rpm at 4 °C for 15 min. The supernatant was kept at −80 °C until virus titer determination. SwIV titers were determined via the Spearman–Kärber method [49,50] in MDCK cells and expressed as tissue culture infectious dose 50% per mL (TCID_50_/mL). PCV2b quantification was performed via qPCR assay as described by Gagnon et al. (2008), using the primers and probe presented in Appendix A [11]. PCV2b concentrations were expressed as DNA copy amount per mL.

### 2.5. Cell Viability Assay

Cell viability assay was performed at 24 h PCV2b/SwIV post-infection with the Celltiter 96 Aqueous One Solution Cell Proliferation Assay kit (Promega, Madison, WI, USA), according to manufacturer’s instruction. Briefly, 20 µL of the reagent were added to each well of the microplate, the cells were then incubated for 2 h at 37 °C and the absorbance was measured at 490 nm (Biotek^®^ Synergy HT plaque reader, Winooski, VT, USA). The infected cells’ viability percentages were calculated using the non-infected cells as control. The experiments were carried out in triplicate and repeated at least two times. 

### 2.6. Cytokine mRNAs Expression in PCV2b- and/or SwIV-Infected Cells

The modulation of mRNAs expression of the following interleukins (ILs), IL-6, IL-8, IL-10 and interferons (IFNs), IFN-α, IFN-β and IFN-γ in co-infected cells versus single-infected cells was assessed via RT-qPCR. Total cellular RNA was extracted from infected and mock-infected NPTr or iPAM 3D4/21 cells and purified with the RNeasy Mini Kit (Qiagen, Valencia, CA, USA), according to the manufacturer’s instructions. Total RNA concentration was measured with a Qubit fluorometer (Thermo Fisher, Walthman, MA, USA). Then, 1 μg of total cellular RNA was reverse transcribed using M-MLV reverse transcriptase (Invitrogen, Burlington, ON, CA), according to the manufacturer’s protocol. After the reverse transcription reaction, the cDNA was used in qPCR reactions with PowerTrack SYBR Green Master Mix kit (Thermo Fisher Scientific, Waltham, MA, USA) on a QuantStudio 3 Real-Time PCR System (Thermo Fisher Scientific). Quantification of differences between groups was carried out using the 2^−ΔΔCt^ method. β_2_-microglobulin (B2M), β-actin (ACTB) and peptidylprolyl isomerase A (PPIA) were used as normalizing genes to compensate for potential differences in cDNA amounts. Mock-infected cells were used as a calibrator reference in the analysis. The primers used for the specific amplification of the targeted cDNA are described in Appendix A [53,54].

### 2.7. 3′mRNA-Seq Library Preparation and Sequencing

Total RNA was purified for RNA-seq analysis from infected cells as described in the previous section. RNA concentration and quality were assessed using Agilent 2100 Bioanalyzer apparatus with the RNA 6000 Nano Kit (Agilent Technologies, Santa Clara, CA, USA). Only RNA samples with acceptable RNA integrity number (RIN > 7) were submitted to library preparation. The cDNA libraries were constructed using Lexogen’s QuantSeq^TM^ 3′mRNA-Seq Kit (Lexogen GmbH, Vienna, Austria) according to the manufacturer’s recommendations. A total of 12 libraries, including 3 libraries from PCV2b-infected cells, 3 from SwIV H1N1-infected cells, 3 from PCV2b/SwIV H1N1-infected cells and 3 from mock-infected cells, were synthesized. Concentration and quality of the purified libraries were assessed using the Qubit fluorometer (Thermo Fisher, Walthman, MA, USA) and an Agilent high-sensitivity DNA kit with a Bioanalyzer (Agilent, CA, USA), respectively. All libraries were sequenced with a MiSeq high-throughput apparatus (Illumina, San Diego, CA, USA) using v3 cartridges (150-cycles) according to the manufacturer’s instructions. 

### 2.8. Bioinformatic Analysis

Raw sequencing data were imported in FASTQ format into CLC Genomics Workbench (version 22.0.1, Qiagen, CA, USA). Reads were trimmed for quality and adaptors using the CLC Genomics Workbench software and trimmed reads were mapped to *Sus scrofa* rRNA 12S, 16S and 18S. After removing the rRNA mapped reads, the remaining unmapped reads were mapped to the reference genome (Sscrofa11.1). The mapped reads were then used for differential expression calculations. Differentially expressed genes (DEGs) with a false discovery rate (FDR) threshold < 0.05 and a fold change >1.5 were selected and used for volcano plots and Venn diagram visualization in CLC Genomics workbench. Pathway enrichment analysis was performed with the Database for Annotation, Visualization and Integrated Discovery (DAVID, 2021) [55,56]. Additionally, the list of DEGs from the co-infected cells was analyzed with ClueGo application (version 2.5.5) available in Cytoscape (version 3.9.1) to identify protein–protein interaction networks and associated cellular pathways [41,42,43,44]. All sequences were deposited in GEO repository (accession number GSE229215). 

### 2.9. Statistical Analyses

All statistical analyses were performed using GraphPad Prism software (GraphPad Prism 7.0.0). The different statistical tests used within each analysis are described in figure legends. Differences were considered significant at *p*-value < 0.05.

## 3. Results

### 3.1. Modulation of Swine Influenza A Virus Replication in Co-Infected PCV2b/SwIV H1N1 Cells

The immunofluorescence assay (IFA) performed on co-infected NPTr and iPAM 3D4/21 cells confirmed the simultaneous presence of PCV2b and SwIV H1N1 antigens in the cells (Figure 1). Similar results were observed in PCV2b/SwIV H3N2 co-infected cells. PCV2b internalized with similar efficiency in NPTr and iPAM 3D4/21 cells as the percentage of PCV2b-positive cells determined via IFA was similar for both types of cells. However, the number of SwIV H1N1-positive cells was significantly higher in infected NPTr cells than in infected iPAM 3D4/21 cells (*p* < 0.001). 

The infectious titer of SwIV H1N1/H3N2 subtypes and the PCV2b viral genome quantity in co-infected cells was determined and compared to single-infected cells at different times post-infection (Figure 2). The SwIV H1N1 and H3N2 infectious titer at 24 h post-infection (hpi) was significantly higher (*p* < 0.0001) in single-infected NPTr cells compared to single-infected iPAM 3D4/21 cells. These results were expected considering that SwIV preferentially infects epithelial cells of the respiratory tract [42]. However, PCV2b qPCR quantification results were similar in both infected NPTr and infected iPAM 3D4/21 cells (Figure 3), confirming the IFA results (Figure 1).

During co-infection, it was found that PCV2b modulated the replication kinetics of both subtypes of SwIV, H1N1 and H3N2, in the infected cells. In fact, the SwIV infectious titer decreased significantly after 24 hpi in PCV2b/SwIV co-infected NPTr cells compared to cells infected only with SwIV (Figure 2A,C). Inversely, in infected iPAM 3D4/21 cells the infectious titers of SwIV H1N1 and H3N2 were significantly higher in co-infected cells versus single-infected cells (Figure 2B,D). However, the presence of SwIV did not seem to influence the PCV2b replication during co-infection. (Figure 3).

### 3.2. Cell Viability Assay

To evaluate whether the modulation of SwIV replication during co-infection could affect cell viability compared to single infection, a cell viability assay was performed at different times post-infection in single-infected and co-infected NPTr and iPAM 3D4/21 cells. As expected, the cell viability decreased from 24 hpi to 72 hpi in co-infected as well as in single-infected cells compared to mock-infected cells (Figure 4). Moreover, the dual infection of PCV2b and SwIV in both NPTr and iPAM 3D4/21 cells significantly decreased the cell viability over time compared to SwIV single-infected cells. Nevertheless, no significant differences were found between PCV2b co-infected and single-infected cells.

### 3.3. Modulation of Cytokines’ mRNAs Expression in Co-Infected Cells

The levels of mRNA expression of pro-inflammatory (IL-6 and IL-8) and anti-inflammatory (IL-10) cytokines, as well as of type I IFN (IFN-α/β) and type II IFN (IFN-γ), were measured at 24 h post-infection in single-infected and co-infected cells. The results obtained with infected NPTr cells revealed that IL-6, IL-8, and IL-10 mRNA expressions were higher in PCV2b/SwIV H1N1 and PCV2b/SwIV H3N2 co-infected cells compared to SwIV H1N1 and SwIV H3N2 single-infected cells, respectively (Figure 5A). However, no difference was found for mRNA expression of interleukins tested between PCV2b/SwIV H1N1- and PCV2b-infected cells. In PCV2b/SwIV H3N2-infected NPTr cells, IL-6 and IL-10, but not IL-8, mRNA expression was reduced compared to PCV2b single-infected cells (Figure 5A).

As shown in Figure 5B, in iPAM 3D4/21 cells, the co-infection with PCV2b and SwIV H1N1 did not significantly impact the IL-6 mRNA expression. However, in co-infected PCV2b/SwIV H3N2 cells, IL-6 expression was reduced compared to SwIV H3N2-infected cells, but not when comparing to PCV2b single-infected cells. IL-10 expression decreased in PCV2b/SwIV H1N1 co-infected iPAM 3D4/21 cells compared to PCV2b- or SwIV H1N1-infected cells. The dual infection of PCV2b and SwIV H3N2 increased the expression of IL-10 regarding to SwIV H3N2 alone, however, the transcriptional level of this cytokine was significantly reduced in these infections compared to PCV2b single-infected cells.

In the case of IFN type I response, a clear synergistic effect was observed in co-infected NPTr cells as IFN-α and IFN-β mRNAs expression was significantly higher in co-infected cells compared to in PCV2b or SwIV (H1N1 or H3N2) single-infected cells (Figure 5C). Similar results were obtained for IFN-γ in PCV2b/SwIV H1N1 co-infected cells compared to single-infected cells. However, although significant differences were observed in the mRNA expression of IFN-γ between PCV2b/SwIV H3N2 and SwIV H3N2 infections, no difference was found when compared to PCV2b single-infected cells (Figure 5C). 

Regarding the IFN response in co-infected iPAM 3D4/21 cells (Figure 5D), no difference was observed on the mRNA expression of the IFNs tested compared to PCV2b-infected cells. Nevertheless, IFN-β and IFN-γ mRNA expressions were significantly higher in SwIV (H1N1 or H3N2) single-infected cells compared to PCV2b/SwIV co-infected cells. Additionally, IFN-α mRNA expression was higher in SwIV H3N2-infected than in co-infected cells. It is interesting to note that the cell type influenced the modulation of IFN mRNA expression. For example, the mRNA expression levels of IFN-α, IFN-β and IFN- γ were up-regulated in NPTr cells following PCV2b infection (Figure 5C) compared to mock-infected cells; however, in PCV2b-infected iPAM 3D4/21, there was a trend toward down-regulation of those mRNAs (Figure 5D). This difference in IFNs’ transcriptional response was also observed between co-infected NPTr and iPAM 3D4/21 cells (Figure 5C,D). Overall, these results at least revealed that dual PCV2b and SwIV infection modulates cytokine transcriptional responses in a cell-type-dependent manner.

### 3.4. Differentially Expressed Genes (DEGs) and Pathway Enrichment Analysis in PCV2b-, SwIV H1N1- and PCV2b/SwIV H1N1-Infected Cells

RNA-seq analysis was performed to characterize and compare the transcriptomic response in NPTr and iPAM 3D4/21 co-infected and single-infected cells. The transcriptomic analyses were performed at 24 hpi to ensure there was a significant amount of suitable viable cells following viral infection. SwIV H1N1 was chosen for the present experiment because during SwIV H1N1/PCV2b co-infection, the modulation of SwIV replication was greater at 24 hpi compared to that with SwIV H3N2/PCV2b co-infection (Figure 2). 

First, 3′mRNA sequencing was used to identify the DEGs in infected cells compared to mock-infected cells. In the NPTr cells, 365, 98 and 627 DEGs were identified in PCV2b-, SwIV H1N1- and PCV2b/SwIV H1N1-infected cells, respectively, when compared to mock-infected cells. (Figure 6A). The number of up-regulated and down-regulated genes in each infection is listed in Appendix A. In iPAM 3D4/21 cells, 316, 64 and 164 DEGs were identified in PCV2b-, SwIV H1N1- and PCV2b/SwIV H1N1-infected cells, respectively, compared to mock-infected cells. (Figure 6B). The number of up-regulated and down-regulated genes in each infection is shown in Appendix A. 

The identified DEGs were used to perform pathway enrichment analysis to determine the cellular pathways impacted during viral infection in single- and co-infected cells. After the enrichment analysis, the top 10 over-represented pathways with an FDR < 0.05 were retained. Only pathways with a fold enrichment > 2 are illustrated in infected NPTr cells (Figure 6). In the case of iPAM 3D4/21 cells, the number of enriched pathways with the same FDR ≥ 2 was extremely high. That is why only pathways with a fold enrichment > 4 are depicted in iPAM 3D4/21 cells. 

The pathway enrichment analysis in NPTr cells revealed the following cellular pathways: ECM–receptor interaction, protein processing in endoplasmic reticulum, focal adhesion and regulation of actin cytoskeleton; these were among the most enriched pathways in PCV2b-infected cells as well as in co-infected cells (Figure 7A,C). Other over-represented pathways in NPTr co-infected cells included mRNA surveillance, HIF-1 signaling pathway, spliceosome, phagosome, NOD-like receptor signaling pathway and influenza A pathway (Figure 7E). In SwIV H1N1-infected NPTr cells, the most enriched pathways were related to the protein synthesis machinery and included the following pathways: translation, ribosome, metabolism of proteins, among others involved in similar functions (Figure 7C). Additionally, nonsense-mediated decay (NMD) pathways were identified in SwIV H1N1-infected NPTr cells. The pathway enrichment analysis performed with iPAM 3D4/21 cells showed that in single-infected as well as in co-infected cells, NF-Kappa B and TNF signaling pathways were the most enriched pathways in the set of DEGs analyzed. However, in co-infected cells, additional enriched cellular pathways were found; these included apoptosis, influenza A, HIF-1 signaling pathway and MAPK signaling pathways (Figure 7B,D–F). 

An additional analysis was performed using the Cytoscape’s ClueGo application with the list of DEGs from the co-infected cells to identify protein–protein interaction networks and to confirm the cellular pathways modulated during PCV2b/SwIV H1N1 co-infection. This analysis confirmed the enrichment of cellular pathways such as mRNA surveillance, spliceosome, protein processing in endoplasmic reticulum, phagosome and regulation of actin cytoskeleton in NPTr co-infected cells (Appendix A). In co-infected iPAM 3D4/21 cells, HIF-1 signaling pathway, TNF signaling, regulation of innate immune response, IL-17 signaling, adaptive immune response based on somatic recombination of immune receptors built from immunoglobulin superfamily domains, apoptosis and influenza A pathways were identified (Appendix A). The Appendix A illustrate several DEGs that are involved in more than one cellular pathway in co-infected NPTr and iPAM 3D4/21 cells, respectively. 

## 4. Discussion

In the present study, PCV2b replication was not modulated in the presence of both SwIV strains (H1N1 or H3N2) nor in both infected cell lines being tested (Figure 3). These results are in accordance with those of previous studies. For example, Wei et al. 2010 concluded that infection of caesarean-derived colostrum-deprived pigs with SIV H1N1, one week after a previous infection with PCV2, did not influence PCV2 replication in dually infected pigs, while PCV2 infection increases SwIV-related clinical disease [39]. Unfortunately, this previous study did not provide any information regarding the effect of PCV2/SwIV co-infection on SwIV replication. Interestingly, our results demonstrated that PCV2b modulated SwIV replication in a cell-type-dependent manner (Figure 2). In fact, PCV2b decreased the replication of SwIV in NPTr cells, whereas the SwIV titer was enhanced in iPAM 3D4/21 co-infected cells. Productive replication of SwIV in macrophages can alter antiviral macrophage functions such as phagocytosis, resulting in enhanced disease severity and impaired bacterial clearance during secondary bacterial infection [47,57]. 

It is noteworthy that at 24 hpi, both PCV2b/SwIV co-infected NPTr and iPAM 3D4/21 cells had a significant decrease in cell viability compared to SwIV single-infected cells. Previous studies have reported that a decreased cell viability significantly reduces the production of infectious viruses from pig respiratory epithelial cells [58]. This would explain why SwIV replication was significantly reduced in PCV2b/SwIV co-infected NPTr cells in the present study. However, in co-infected iPAM 3D4/21 cells, the SwIV replication was rather enhanced, even when the cell viability was decreasing over time in dual-infected cells compared to SwIV single-infected cells. Previous studies revealed that PCV2b infection in PAMs activates NF-κB, phospho-Akt and MAPK signaling pathways in infected cells. The activation of these signaling pathways can be detected in the first 30 min after PCV2 infection [59]. It is known that influenza virus replication starts early after virus entry in the cells. Viral mRNA starts to accumulate within the first hour post-infection and viral genome replication starts 1.5 to 2 hpi [60]. The modulation of cytokines and the activation of NF-κB and MAPK signaling pathways, among others, have been shown to be important for a successful influenza virus infection [61]. It is possible that the modulation of cytokines’ expression and the activation of signaling pathways in iPAM 3D4/21 cells in the first 24 h after PCV2b infection could lead to an enhanced SwIV replication in co-infected cells in the present study, even when the cell viability decreased throughout the viral infection.

PCV2b has been previously reported to induce up-regulation of cytokines such as IL-6, IL-8 and IL-10 in vitro [59,62,63]. In the present study, IL-6 and IL-10 mRNA expressions were increased via PCV2b infection in NPTr and iPAM 3D4/21 cells in accordance with previous studies reporting up-regulation of IL-6 in epithelial cells [63] and increased levels of IL-6 and IL-10 in porcine alveolar macrophages (PAMs) [64,65]. In the present study, IL-8 mRNA was upregulated in PCV2b-infected NPTr cells, as expected, whereas it was not in PCV2b-infected iPAM 3D4/21 cells (Figure 5). These results do not correlate with those of previous reports showing a significant up-regulation of IL-8 mRNA expression and/or protein production in PCV2-inoculated swine alveolar macrophages [66,67]. The contradictory results obtained in the present study may be due to the time points selected to measure the expression of this cytokine in macrophages after PCV2b infection or the use of an immortalized PAM cell line (3D4/21) instead of primary porcine alveolar macrophages. It is noteworthy that cytokine mRNA expression was determined in persistently PCV2b-infected cells to mimic the superinfection events that occurs in PCV2b previously infected pigs. Therefore, a persistent PCV2b infection in iPAM 3D4/21 cells could have a differential effect on mRNA expression compared to an acute PCV2b cell infection. 

Induction of pro-inflammatory cytokines during SwIV infection has been correlated with viral replication and clinical signs [68]. In the current study, IL-6, IL-8, and IL-10 mRNA expressions were higher in PCV2b/SwIV (H1N1 or H3N2) co-infected NPTr cells compared to SwIV H1N1 and SwIV H3N2 single-infected cells. However, no difference was observed regarding the tested interleukin mRNA expression levels between PCV2b/SwIV H1N1- and PCV2b-infected cells. Nevertheless, in PCV2b/SwIV H3N2-infected cells, IL-6 and especially IL-10 mRNA expression levels were significantly reduced compared to PCV2b single-infected NPTr cells (Figure 5A). These results reveal that the modulation of interleukin expression via PCV2b/SwIV co-infection on NPTr cells might be influenced by the SwIV genotype being investigated.

Regarding the mRNA expression of antiviral cytokines such as IFNs, their expression level modulations in co-infected cells were found to be cell-type-dependent. In NPTr cells, the IFN type I (α/β) mRNA expression levels were synergistically up-regulated in co-infected cells compared to single-infected cells, whereas in iPAM 3D4/21 cells, a trend towards a reduced IFN type I and II mRNA expression was observed in PCV2b single-infected and co-infected cells (Figure 5). Several researchers have previously reported contradictory data in regard to the modulation of IFN response following infection with PCV2b. Wang et al. (2022) reported that PCV2 infection interferes with the activation of type I IFNs signaling pathway and inhibits the IFN-induced ISGs expression in vivo and in vitro [69]. Others have reported the inhibition of IFN-α in porcine peripheral blood mononuclear cells (PBMCs) [70,71] and the inhibition of type I IFN (IFN-α /IFN-β) in PK-15 cells [72,73] following PCV2 infection or in the presence of PCV2 viral protein or DNA. In addition, according to Gao et al. (2014), PCV2 significantly inhibited pseudorabies virus (PRV)-induced IFN-γ mRNA expression in swine PBMC in vitro [74]. However, other authors have reported that porcine circovirus type 2 induces type I interferon production in porcine alveolar macrophages [75] and IFN-β in PK-15-infected cells [76]. Interestingly, it is known that the PCV2 genome contains CpG motifs with both IFN-α inhibitory and stimulatory properties [77]. Kekarainen et al. (2008b) suggested that PCV2 viral elements can distinctly regulate cytokine production according to the cell population [78]. Overall, it is easy to conclude that IFN response modulation following PCV2 infection is complex and could vary depending on the experimental models being studied (in the present study: tracheal epithelial cells versus macrophages). 

It is known that type I and III IFNs are rapidly induced after influenza virus sensing via pattern recognition receptors (Toll-like receptors, RIG-I-like receptors, NOD-like receptors) on respiratory epithelial cells. The released IFNs bind to their cognate receptors and activate the JAK-STAT signaling pathway. This triggers the transcription of hundreds of IFN-stimulatory genes (ISGs) that induce an “antiviral state” in infected and nearby cells to ultimately restrict viral replication and propagation [79,80,81,82,83]. In a study performed by Wu et al. (2022), the authors overexpressed microRNA let-7 in A549 cells, which resulted in an increase of interferon type I mRNA expression and consequently inhibited influenza virus infection [84]. Moreover, Fong et al. (2022) demonstrated that IFN-γ inhibits influenza virus replication in respiratory epithelial cells by reducing viral binding onto the cells [85]. The results obtained during the present study suggest that the increased expression of IFN mRNA in PCV2b/SwIV-infected NPTr cells could lead to a reduction in SwIV viral replication compared to single SwIV-infected cells. It is also well-known that the influenza virus possesses several strategies to counteract IFN response [86,87]. That would explain why the increased IFN mRNA expression in co-infected NPTr cells could be responsible for a partial reduction in, and not a complete shutdown of, SwIV replication (Figure 2). Interestingly, Czerkies et al. (2022) have reported interference between respiratory syncytial virus (RSV) and influenza A virus H1N1 in human alveolar epithelial cells (A549). The authors of the report found that previous infection of cells with RSV does not prevent a subsequent influenza virus infection, whereas RSV indeed protects bystander cells against influenza virus infection by triggering secretion of type I and type III IFNs [88]. A similar effect in PCV2b/SwIV co-infected NPTr cells could be involved and explain the PCV2b impact on the apparent inhibition of SwIV replication. However, this PCV2b bystander phenomenon effect on SwIV replication is highly plausible, though it still needs to be demonstrated. In iPAM 3D4/21 co-infected cells, PCV2b completely impaired the modulation of the transcriptional expression of IFN-β and IFN-γ by SwIV. Gao et al. (2014) have reported a similar PCV2 effect on IFN expression during dual infection with pseudorabies virus (PRV) in PBMC [74]. In this previous study, PCV2 significantly inhibited the ability of inactivated PRV to induce IFN-γ expression. The authors suggested that PCV2 could affect the cell immune response to PRV [74]. In the context of the present study, it is believed that the impaired IFN response observed in PCV2b persistently infected iPAM 3D4/21 cells could favor SwIV infection and replication. It is important to point out that although a modulation of the expression of IFNs’ mRNA was revealed in the infected cells, these results require confirmation using other methods, such as ELISA.

Differential transcriptome analysis of PCV2b/SwIV H1N1 and single-infected porcine cells revealed distinct signatures on host gene expression that may have impacted SwIV H1N1 replication during the co-infection. Several of the most enriched pathways identified in co-infected cells in this study are known to be involved in the host antiviral response and influenza virus pathogenesis and propagation [61] such as NOD-like receptor signaling pathway in co-infected NPTr cells and apoptosis and MAPK signaling pathways in co-infected iPAM 3D4/21 cells (Figure 7). NOD-like receptors (NLR) are a specific family of cytosolic pattern-recognition receptors (PRRs) that contains more than 20 members in mammals and plays a pivotal role in the recognition of intracellular ligands [89]. A group of NLRs are associated with the formation of the inflammasome, a cytosolic multiprotein complex that, once activated, induces the production of IL-1β and IL-18 and the recruitment of immune cells to the site of infection [90,91]. The NLRP3 inflammasome has an important role in the immune response against influenza A virus infection. Its role in both protective and detrimental immune responses during influenza A virus infection has been studied and extensively reviewed [91,92,93,94]. Interestingly, NLRP3 was found to be up-regulated in PCV2b/SwIV H1N1-infected NPTr cells (FC 6.16, FDR = 0.04) but not in SwIV H1N1 and PCV2b single-infected cells. Moreover, DDX3X, a host protein which has been implicated in the coordination of host defense against influenza virus by activating the NLRP3 inflammasome and type I interferon response [95,96] was upregulated in co-infected NPTr cells (FC 3, FDR = 5.84 × 10^−4^) and in PCV2b-infected cells (FC 2.42, FDR = 0.01), but not in SwIV H1N1-infected cells.

Apoptosis and MAPK pathways can be modulated during the replication cycle of the influenza virus to favor viral infection. Wurzer et al. (2003) have found that caspase 3 activation during the onset of apoptosis is important for efficient influenza virus propagation [97]. The MAPK pathway, on the other hand, has been shown to positively regulate multiple steps of influenza virus replication such as RNA synthesis, vRNP export and release (budding) of virus, clathrin-independent endocytosis and viral internalization, nuclear export and protein synthesis, (see the review of Kumar et al., 2018 [98] for information and references regarding the role of MAPK signaling in the replication of several virus, including influenza virus).

HIF-1 signaling pathway was identified among the most modulated pathways in the co-infected NPTr and iPAM 3D4/21 cells but not in PCV2b or SwIV H1N1 single-infected cells (Figure 7). HIF-1-related pathways have been reported to be involved in viral infections and in the innate and the adaptive immune responses of the host against viruses [99]. Previous reports regarding HIF-1’s role in influenza virus pathogenesis are contradictory. It was reported that the H1N1 virus activates HIF-1 pathway and then induces glycolysis to promote viral replication, whereas knockdown of this pathway significantly reduced H1N1 replication in A549 cells [100]. However, in another study, it was demonstrated that HIF-1α knockdown in A549 cells promoted influenza A virus replication by promoting autophagy in cells [101].

## 5. Conclusions

In conclusion, it was found that PCV2b decreases SwIV replication in porcine tracheal epithelial cells while enhancing SwIV replication in alveolar porcine macrophages, probably through modulation of IFN expression. In addition, PCV2b and SwIV were found to synergistically enhance IFNs mRNA expression in infected NPTr cells, whereas in iPAM 3D4/21 cells, PCV2b impaired the capacity of SwIV to promote IFNs mRNA expression. Cellular genes and pathways were found to be differentially modulated in PCV2b/SwIV H1N1 co-infected cells compared to PCV2b and SwIV H1N1 single-infected cells. However, further studies are needed to establish the role of identified DEGs and cellular pathways in the pathogenesis of PCV2b/SwIV co-infection. 

## Figures and Tables

**Figure 1 viruses-15-01207-f001:**
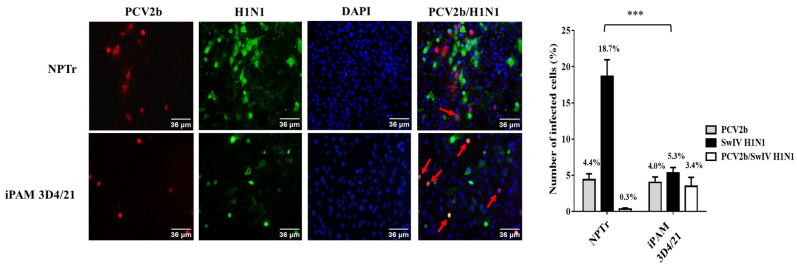
PCV2b and SwIV H1N1 co-localization in both NPTr and iPAM 3D4/21 co-infected cells. IFA was performed at 24 h post-infection to detect PCV2b (red) and SwIV H1N1 (green) in co-infected cells. The cells were infected with 0.05 MOI of PCV2b and passaged 2–3 times. Then, PCV2b-infected cells were infected with 1 MOI of SwIV H1N1. Nuclear staining with DAPI is shown in blue. Co-localization pictures were realized with ImageJ. The red arrows point to single cells expressing both PCV2b and SwIV H1N1 antigens. Statistical analyses were carried out using a two-way ANOVA followed by Sidak’s multiple comparison test (GraphPad Prism software, version 7.00). *** *p* < 0.001.

**Figure 2 viruses-15-01207-f002:**
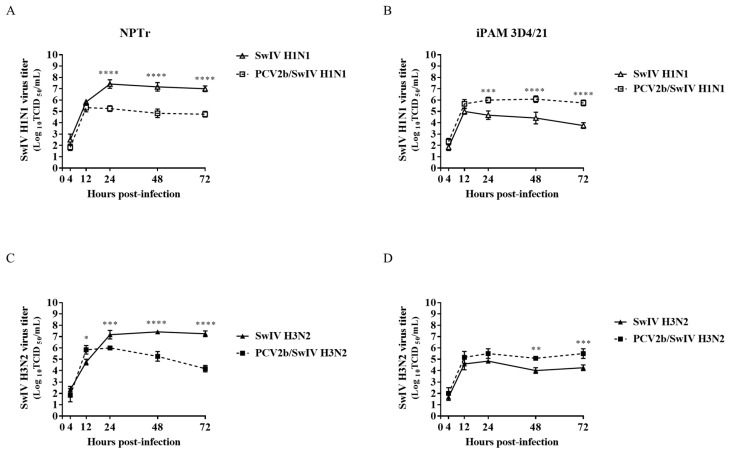
PCV2b co-infection effects on SwIV replication in infected NPTr and iPAM 3D4/21 cells. The SwIV titer was determined in MDCK cells using the Spearman–Kärber method and expressed in tissue culture infectious dose 50% per mL (TCID_50_/mL). The PCV2b/SwIV H1N1 (**A**) and PCV2b/SwIV H3N2 (**C**) co-infection experiment in NPTr cells, and PCV2b/SwIV H1N1 (**B**) and PCV2b/SwIV H3N2 (**D**) co-infection experiment in iPAM 3D4/21 cells were repeated 3 times. Data are presented with standard deviation (SD) values. Statistical analyses were carried out using two-way repeated-measures ANOVA (GraphPad Prism software, version 7.00). * *p* < 0.05, ** *p* < 0.01, *** *p* < 0.001, **** *p* < 0.0001.

**Figure 3 viruses-15-01207-f003:**
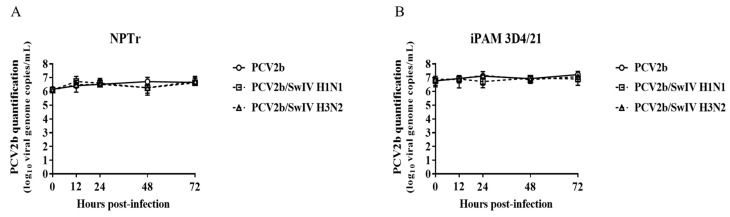
SwIV co-infection effects on PCV2b replication in infected NPTr and iPAM 3D4/21 cells. PCV2b quantification in co-infected NPTr (**A**) and iPAM 3D4/21 (**B**) cells was performed using qPCR assay and expressed as PCV2b genome copies per mL of sample. The experiments were repeated 3 times. Mock-infected cells were PCV2 qPCR-negative (with Ct > 36) and the results are not illustrated in the figure. Data are presented with standard deviation (SD) values. Statistical analyses were carried out using two-way repeated-measures ANOVA (GraphPad Prism software, version 7.0.0).

**Figure 4 viruses-15-01207-f004:**
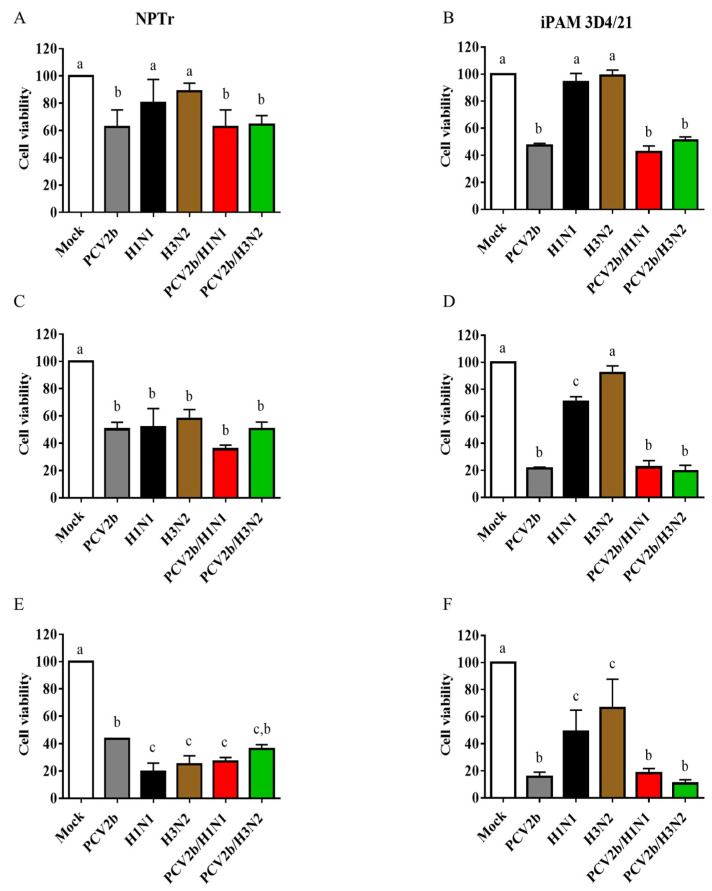
Cell viability was determined in single-infected and co-infected PCV2b/SwIV cells at 24 hpi (**A**,**B**), 48 hpi (**C**,**D**) and 72 hpi (**E**,**F**) in NPTr (**A**,**C**,**E**) and iPAM 3D4/21 cells (**B**,**D**,**F**). The experiments were repeated three times. The data represent percentage of cell viability in infected cells with respect to mock-infected cells and are presented with standard deviation (SD) values. Statistical analyses were carried out using one-way ANOVA followed by Tukey’s multiple comparison test (GraphPad Prism software, version 7.0.0). Different superscripts indicate a statistically significant difference (*p* < 0.05) between groups.

**Figure 5 viruses-15-01207-f005:**
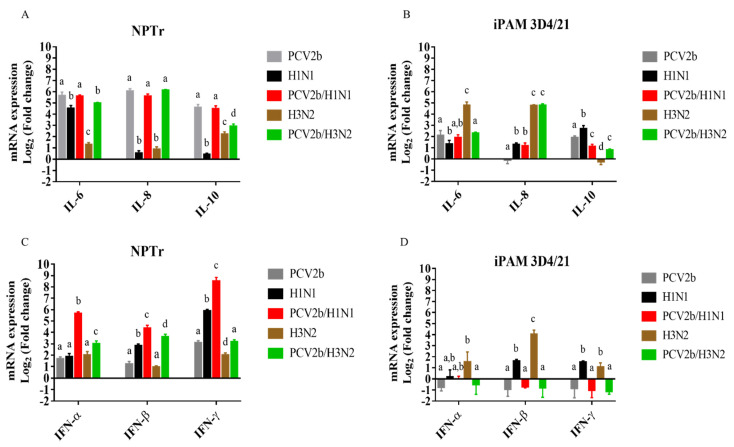
PCV2b/SwIV co-infection effects on the modulation of cytokines’ mRNA expression. The mRNA expressions of IL-6, IL-8 and IL-10 in NPTr cells (**A**) and iPAM 3D4/21 (**B**) as well as of IFN-α, IFN-β and IFN-γ in NPTr cells (**C**) and iPAM 3D4/21 (**D**) were determined using RT-qPCR assays. The 2^−ΔΔCt^ method was used to calculate the fold change of cytokine mRNA expression in infected cells with respect to mock-infected cells at 24 h post-infection. The experiments were repeated at least three times. All data are presented with standard deviation (SD) values. Statistical analyses were carried out using an ordinary two-way ANOVA followed by Tukey’s multiple comparison test (GraphPad Prism software, version 7.0.0). Different superscripts indicate a statistically significant difference (*p* < 0.01) between groups within the same tested cytokine mRNA.

**Figure 6 viruses-15-01207-f006:**
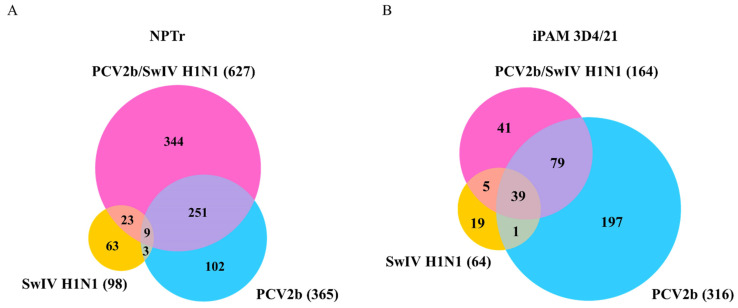
Venn diagram of differentially expressed genes (DEGs) in PCV2b/SwIV co-infected cells. Mock-infected cells were used as control to identify DEGs in NPTr (**A**) and iPAM 3D4/21 (**B**) single-infected and co-infected cells. The numbers in overlapping areas represent the number of DEGs shared among the different virus infection experimental groups. A false discovery rate (FDR) < 0.05 and a fold change cut-off of 1.5 was used to identify the DEGs.

**Figure 7 viruses-15-01207-f007:**
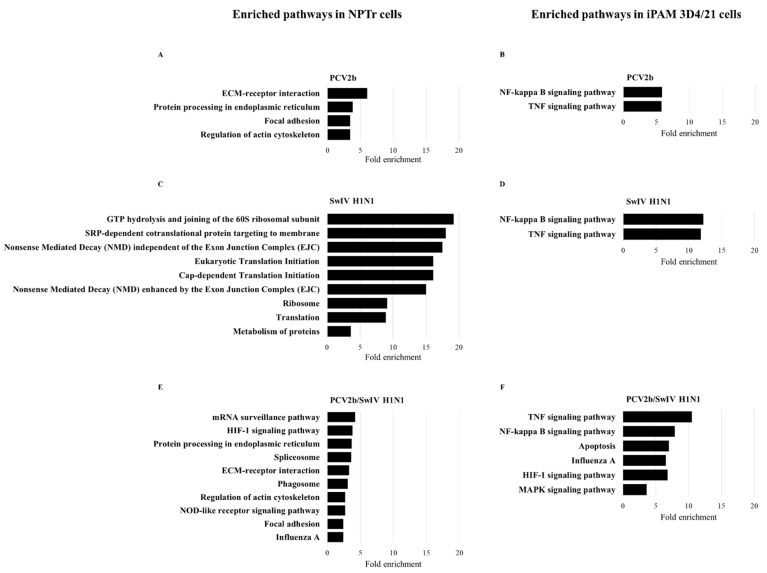
KEGG pathway enrichment analysis of the DEGs identified in single-infected and co-infected cells. Enriched KEGG pathways after PCV2b, SwIV H1N1 and PCV2b/SwIV H1N1 infection in NPTr cells ((**A**,**C**,**E**), respectively) and in iPAM 3D4/21 cells ((**B**,**D**,**F**), respectively) are illustrated. The FDR value < 0.05 was considered for statistically significant enriched pathways.

## Data Availability

The data presented in this study are openly available in Preprints.org (accessed on 18 May 2023) at DOI: 10.20944/preprints202304.0781.v1, reference number preprints-71630.

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
