# Peer review of "Porcine Circovirus Modulates Swine Influenza Virus Replication in Pig Tracheal Epithelial Cells and Porcine Alveolar Macrophages"

_viruses, 2023, doi:10.3390/v15051207_

Round 1

Reviewer 1 Report

In this manuscript, authors demonstrated that PCV2b decreases SwIV replication in porcine tracheal epithelial cells while improving SwIV replication in alveolar porcine macrophages, compared to single infected cells. Besides, PCV2b/SwIV co-infection synergistically up-regulated IFN expression in NPTr cells whereas PCV2b impaired the SwIV IFN induced response in iPAM 3D4/21 cells. Also, RNA-sequencing analyses revealed that the modulation of cellular genes and pathways during PCV2b/SwIV H1N1 co-infection. In general, this manuscript has an important reference for readers. I suggest the editor considers the Manuscript for publication. However, I have seral comments listed as below:
1. Please add scale bar in Figure 1. Also, add a result of normal cells to verify the specificity of the antibody.

2. Please provide the sequences information (like GenBank number) of these strains (PCV2b strain FMV-06-0732, SwIV H1N1 A/swine/St-Hyacinthe/148/1990 and the SwIV H3N2 A/swine/Quebec/1708732/2015) in this study.

3. Add a result of normal cells in Figure 3 to verify the specificity of primers/probe of PCV2b.

4. Although the author mentions that no significant differences were found between PCV2b co-infected and single infected cells, there is a significant difference between SwIV co-infected and single infected cells (Figure 4). Please discuss whether it will affect the analysis of subsequent results.

5. The presentation of this section "3.3. Modulation of cytokines mRNAs expression in co-infected cells " should be re-organized, and it should be more logical.

Reviewer 2 Report

Comments for the author of Viruses manuscript viruses-2386253:

The author of the Viruses manuscript “Porcine circovirus modulates swine influenza virus replication in pig tracheal epithelial cells and porcine alveolar macrophages”, present their findings toward understanding co-infections with swine influenza viruses and porcine circovirus type 2b (PCV2b).  To study these interactions, they are using newborn porcine tracheal epithelial cells (NPTr) and immortalized porcine alveolar macrophages (iPAM 3D4/21).  The readouts are viral replication, cell viability, and cytokine mRNA expression.  Their data show that PCV2b decreased and improved SwIV replication in co-infected NPTr and iPAM3D4/21 cells, respectively.  The co-infection yielded increases in IFN responses from epithelial cells (which reduced virus replication) and decreases in IFN from macrophages (increasing virus replication).  RNA-sequencing shows that modulation of gene expression is in a cell type-dependent manned for PCV2b/SwIV H1N1 co-infection.  This work is of interest and timely as we continue to understand how co-infections impact potential disease severity.  While the study is of interest, below are some comments that I would like the authors to address as they revise the manuscript.   

General Comments:

  1. In the results presented in Figure 4, it seems that the cell viability matches with the PCV2b for both the NPTr and iPAM 3D4/21 cells.  I was wondering if the authors are able to expand on the impact of cell viability as it relates to virus titer.
  2. The paragraph on page 13 is extremely long.  The authors may what to consider breaking that into at least 2 paragraphs.
  3. While the authors show that IFNs are differentially modulated in this study, there are no mechanistic studies to fully support this conclusion. 
